# Streamlines of the Poynting Vector and Chirality Flux around a Plasmonic Bowtie Nanoantenna

**DOI:** 10.3390/nano14010061

**Published:** 2023-12-25

**Authors:** Yun-Cheng Ku, Mao-Kuen Kuo, Jiunn-Woei Liaw

**Affiliations:** 1Department of Mechanical Engineering, Chang Gung University, 259 Wen-Hwa 1st Rd., Kwei-Shan, Taoyuan 333, Taiwan; d05543005@ntu.edu.tw; 2Institute of Applied Mechanics, National Taiwan University, 1, Sec. 4, Roosevelt Rd., Taipei 106, Taiwan; 3Department of Mechanical Engineering, Ming Chi University of Technology, New Taipei City 24301, Taiwan; 4Proton and Radiation Therapy Center, Linkou Chang Gung Memorial Hospital, Taoyuan 333423, Taiwan

**Keywords:** Poynting vector, nanocube, bowtie nanoantenna, nanotriangle, chirality flux, plasmon, Stratton–Chu formulation, surface integral equations, boundary element method, method of moments, photocatalysis

## Abstract

The streamlines of the energy flux (Poynting vectors) and chirality flux as well as the intensity of the electric field around various plasmonic nanostructures (nanocube, nanocuboid, nanotriangle, hexagonal nanoplate and bowtie nanoantenna) induced by a circularly polarized (CP) or linearly polarized (LP) light were studied theoretically. The boundary element method combined with the method of moment was used to solve a set of surface integral equations, based on the Stratton–Chu formulation, for analyzing the highly distorted electromagnetic (EM) field in the proximity of these nanostructures. We discovered that the winding behavior of these streamlines exhibits versatility for various modes of the surface plasmon resonance of different nanostructures. Recently, using plasmonic nanostructures to facilitate a photochemical reaction has gained significant attention, where the hot carriers (electrons) play important roles. Our findings reveal a connection between the flow pattern of energy flux and the morphology of the photochemical deposition around various plasmonic nanostructures irradiated by a CP light. For example, numerical results exhibit vertically helical streamlines of the Poynting vector around an Au nanocube and transversely twisted-roll streamlines around a nanocuboid. Additionally, the behaviors of the winding energy and chirality fluxes at the gap and corners of a plasmonic bowtie nanoantenna, implying a highly twisted EM field, depend on the polarization of the incident LP light. Our analysis of the streamlines of the Poynting vector and chirality flux offers an insight into the formation of plasmon-enhanced photocatalysis.

## 1. Introduction

Light–matter interaction has always been an interesting research topic in the past decades. In particular, the plasmon-enhanced photochemical reactions involving hot carriers (electrons) have been paid attention [1,2,3,4,5,6,7,8,9]. For example, as a gold nanocube immersed in an aqueous solution of Pb(NO_3_)_2_ is irradiated by a broad-band light with wavelengths longer than 520 nm, a helical dielectric cap of PbO_2_ grows along the surface of the nanocube to form a chiral composite nanospiroid [1]. In this process, plasmon-enhanced photo-oxidation takes place [1,2]. To explain these experiments, a theory of plasmon-induced charge separation (PICS) was proposed [1,2]. In addition, as an Ag nanotriangle was irradiated by an 810 nm laser in an aqueous solution of AuHCl_4_, the photo-reduction of Au clusters deposited locally on the corners was observed [3]. Refs. [3,4,5] showed the photochemical synthesis for growing gold nanoprisms (nanotriangle and hexagonal nanoplate) within an aqueous solution of HAuCl_4_. Moreover, when a 4-NTP-coated Ag bowtie antenna is exposed to a 633 nm laser beam, it leads to the generation of hot electrons from the bowtie antenna, initiating the reduction in 4-NTP and, subsequently, its conversion into 4-ATP [6]. In this experiment, the photo-reduction predominantly takes place at the gap and edges of the bowtie antenna [6]. The plausible explanation is that local electric-field hotspots, typically found in the vicinity of the gap, are the primary sites for photocatalysis [6]. The local rate of hot-electron generation was used to explain the growth of photochemical deposition [10,11,12]. Furthermore, the highly winding streamlines of energy flux (Poynting vector) around certain hotspots were also proposed to explain the phenomenon in a previous study [13]. In addition, the role of reactive energy flux (power flow) in the near field is also important [14]. For the Poynting vector, the concepts of optical chirality and chirality flux were proposed to interpret the locally distorted electromagnetic (EM) field [15,16,17,18]. To further study these quantities with the phenomena of a photochemical reaction, several simulation methods, e.g., the finite difference time domain method, boundary element method (BEM) [13,19,20,21,22], method of moment (MoM) [23], finite element method (FEM) [4,12,24], multiple multipole (MMP) method [25], etc., have been used to investigate the local hotspots of an electric field on plasmonic nanostructures at different plasmon modes [26]. In particular, FEM (COMSOL) was used to study the incremental process of hot-electron-induced photogrowth [12]. In addition, a variety of applications of a plasmonic bowtie nanoantenna, e.g., photothermal, photochemical, Fano resonance for sensing and optical trapping, have been studied recently [9,27,28,29,30,31,32,33,34,35,36].

To realize the plasmon-enhanced photoredox, we are motivated to study the streamlines of the Poynting vector (energy flux) [13] and chirality flux [15] as well as the intensity of the electric field around a plasmonic nanostructure, particularly a bowtie nanoantenna. In this paper, a set of surface integral equations (SIEs) of Fredholm equations of the second kind, based on the Stratton–Chu formulation, was derived for the EM-field simulation [13,37,38,39,40]. Four coupled SIEs are in terms of the surface electric/magnetic charges and currents as unknowns, where the kernel functions are the combinations of two scalar Green’s functions for the interior and exterior media [13]. We adopt BEM combined with MoM to simulate the EM field for the study of the surface plasmon resonance (SPR) of various Au nanostructures and the impact on a plasmon-enhanced photochemical reaction [13]. In particular, the streamlines of the Poynting vector and chirality flux are analyzed to manifest the winding behavior around various plasmonic nanostructures (e.g., nanocube or bowtie nanoantenna), irradiated by a circularly polarized (CP) plane wave. Moreover, the polarization-dependent performance of an Ag bowtie nanoantenna irradiated by a linearly polarized (LP) light is also discussed.

## 2. Method

We use BEM combined with MoM to deal with a set of four coupled SIEs derived from the Stratton–Chu formulation [13]. In these SIEs, there are four sets of physical unknowns: the surface electric/magnetic charges and currents [13,37]. In particular, the RWG algorithm is used to model the surface electric and magnetic currents on the surface of a multi-connected scatterer (e.g., bowtie nanoantenna) [13,41]. Figure 1 demonstrates the implementation of BEM/MoM with the discretization of the surface of a multi-connected scatterer, e.g., a bowtie nanoantenna. Subsequently, a system of algebraic equations is constructed for us to solve these surface unknowns. After these surface unknowns are obtained, the EM field in the surrounding medium can be obtained using the surface integral representations (SIRs) of an EM field in the exterior or interior domain, and then the distributions of the streamlines of the Poynting vector and chirality flux around the nanostructure can be analyzed. The theoretical method is illustrated in detail as follows.

Throughout this paper, the time harmonic factor is e*^−jωt^*. According to the Stratton–Chu formulation, the EM fields at a position vector **r**′ in the interior region with a surface *S* of the scatterer (i.e., nanostructure) can be expressed by the SIRs as
(1)$$E_{1}\left(r^{\prime }\right)=\int_{S}-j\omega \mu_{1}G_{1}\left(r,r^{\prime }\right)J\left(r\right)-\nabla G_{1}\left(r,r^{\prime }\right)\frac{D_{n}\left(r\right)}{\varepsilon_{1}}-\nabla G_{1}\left(r,r^{\prime }\right)\times M\left(r\right)ds$$
(2)$$H_{1}\left(r^{\prime }\right)=\int_{S}-j\omega \varepsilon_{1}G_{1}\left(r,r^{\prime }\right)M\left(r\right)-\nabla G_{1}\left(r,r^{\prime }\right)\frac{B_{n}\left(r\right)}{\mu_{1}}+\nabla G_{1}\left(r,r^{\prime }\right)\times J\left(r\right)ds$$
(3)$$E_{2}\left(r^{\prime }\right)=E^{i}\left(r^{\prime }\right)+\int_{S}j\omega \mu_{2}G_{2}\left(r,r^{\prime }\right)J\left(r\right)+\nabla G_{2}\left(r,r^{\prime }\right)\frac{D_{n}\left(r\right)}{\varepsilon_{2}}+\nabla G_{2}\left(r,r^{\prime }\right)\times M\left(r\right)ds$$
(4)$$H_{2}\left(r\prime \right)=H^{i}\left(r\prime \right)+\int_{S}j\omega \varepsilon_{2}G_{2}\left(r,r^{\prime }\right)M\left(r\right)+\nabla G_{2}\left(r,r^{\prime }\right)\frac{B_{n}\left(r\right)}{\mu_{2}}-\nabla G_{2}\left(r,r^{\prime }\right)\times J\left(r\right)ds$$
where **J** and **M** are the surface electric and magnetic currents, respectively, and *D_n_* and *B_n_* are the surface electric and magnetic charges, respectively; **J** = **e***_n_* × **H**, **M** = −**e***_n_* × **E**, *D_n_* = **e***_n_*·*ε*_1_**E** and *B_n_* = **e***_n_*·*μ*_1_**H [13]**. Here, **e***_n_* is the outward unit normal vector from the surface of the scatterer. In Equations (1)–(4), the *ε_α_* and *μ_α_* are the permittivity and permeability; *α* = 1 and 2 for the scatterer and the surrounding medium, respectively. The scalar Green’s function *G_α_* (*α* = 1 or 2) satisfies the scalar Helmholtz equation in the scatterer or surrounding medium:(5)$$G_{\alpha }(r,r\prime )=\frac{e^{jk_{\alpha }\left|r-r\prime \right|}}{4\pi \left|r-r^{\prime }\right|}$$
where $k_{\alpha }=\omega \sqrt{\varepsilon_{\alpha }\mu_{\alpha }}$. Three types of surface integral operators in terms of Green’s function *G_α_* are defined as
(6)$$L_{\alpha }\left(r\prime ,J\right)=\int_{S}-j\omega \mu_{\alpha }G_{\alpha }\left(r,r^{\prime }\right)J\left(r\right)ds$$
(7)$$Q_{\alpha }\left(r\prime ,D_{n}\right)=\int_{S}-\nabla G_{\alpha }\left(r,r^{\prime }\right)\frac{D_{n}\left(r\right)}{\varepsilon_{\alpha }}ds$$
(8)$$K_{\alpha }\left(r\prime ,M\right)=\int_{S}-\nabla G_{\alpha }\left(r,r^{\prime }\right)\times M\left(r\right)ds$$
where the subscript “*α*” represents the medium index (1 or 2). Based on the continuity conditions of *D_n_*, *B_n_*, **E***_t_* and **H***_t_* at the interface in terms of the aspects of the interior and exterior fields, four coupled SIEs are obtained [13]:(9)$$\left[\begin{array}{cccc} I+e_{n^{\prime }}\times \left(K_{1}-K_{2}\right) & e_{n^{\prime }}\times \left(L_{1}-L_{2}\right) & e_{n^{\prime }}\times \left(Q_{1}-Q_{2}\right) & 0\\ -e_{n^{\prime }}\times \left(\frac{\varepsilon_{1}}{\mu_{1}}L_{1}-\frac{\varepsilon_{2}}{\mu_{2}}L_{2}\right) & I+e_{n^{\prime }}\times \left(K_{1}-K_{2}\right) & 0 & -e_{n^{\prime }}\times \left(\frac{\varepsilon_{1}}{\mu_{1}}Q_{2}-\frac{\varepsilon_{2}}{\mu_{2}}Q_{2}\right)\\ -e_{n^{\prime }}\cdot \left(\varepsilon_{1}K_{1}-\varepsilon_{2}K_{2}\right) & -e_{n^{\prime }}\cdot \left(\varepsilon_{1}L_{1}-\varepsilon_{2}L_{2}\right) & 1-e_{n^{\prime }}\cdot \left(\varepsilon_{1}Q_{1}-\varepsilon_{2}Q_{2}\right) & 0\\ -e_{n^{\prime }}\cdot \left(\varepsilon_{1}L_{1}-\varepsilon_{2}L_{2}\right) & e_{n^{\prime }}\cdot \left(\mu_{1}K_{1}-\mu_{2}K_{2}\right) & 0 & 1-e_{n^{\prime }}\cdot \left(\varepsilon_{1}Q_{1}-\varepsilon_{2}Q_{2}\right) \end{array}\right]\left[\begin{array}{c} M\\ J\\ D_{n}\\ B_{n} \end{array}\right]=\left[\begin{array}{c} -e_{n^{\prime }}\times E^{i}\\ e_{n^{\prime }}\times H^{i}\\ e_{n^{\prime }}\cdot \varepsilon_{2}E^{i}\\ e_{n^{\prime }}\cdot \mu_{2}H^{i} \end{array}\right]$$
where the superscript ‘*i*’ denotes the incident field in the surrounding medium. In the above system of algebraic equations, *D_n_*, *B_n_*, **J** and **M** are the surface unknowns for each discretized mesh, and **I** is the identity matrix of the order of 3 × 3. Through the discretization of the surface of the interface, a system of algebraic equations of the coupled SIEs can be constructed using BEM and MoM [13]. Consequently, the surface unknowns are solved from the system of algebraic equations. Subsequently, the EM fields at any position vector **r**′ in the surrounding medium can be calculated in terms of this surface information by the following SIRs in terms of the surface information of *D_n_*, *B_n_*, **J** and **M**:(10)$$E_{1}\left(r\prime \right)=L_{1}\left(r\prime ,J\right)+Q_{1}\left(r\prime ,D_{n}\right)+K_{1}\left(r\prime ,M\right)$$
(11)$$H_{1}\left(r\prime \right)=\frac{\varepsilon_{1}}{\mu_{1}}L_{1}\left(r\prime ,M\right)+\frac{\varepsilon_{1}}{\mu_{1}}Q_{1}\left(r\prime ,B_{n}\right)-K_{1}\left(r\prime ,J\right)$$

Similarly, the EM fields at any position vector **r**′ in the scatterer can be calculated by
(12)$$E_{2}\left(r\prime \right)=E^{i}\left(r\prime \right)-L_{2}\left(r\prime ,J\right)-Q_{2}\left(r\prime ,D_{n}\right)-K_{2}\left(r\prime ,M\right)$$
(13)$$H_{2}\left(r\prime \right)=H^{i}\left(r\prime \right)-\frac{\varepsilon_{2}}{\mu_{2}}L_{1}\left(r\prime ,M\right)-\frac{\varepsilon_{2}}{\mu_{2}}Q_{2}\left(r\prime ,B_{n}\right)+K_{2}\left(r\prime ,J\right)$$

In the following numerical analysis, we use the above method to calculate the EM fields in the surrounding medium first and then to simulate the energy flux (Poynting vector) **S***_avg_* [13],
(14)Savg=12Re⁡E×H¯
and the chirality flux **F** [15,16],
(15)$$F=\frac{j\omega }{4}\left(E\times \bar{D}-\bar{H}\times B\right)$$

Throughout this paper, the bar is the conjugate. In addition, the chirality density *C* is defined as [16]
(16)C=ω2Im⁡D⋅B¯

The relationship between the chirality density and flux is
(17)−ω24Re⁡D⋅μH¯−ε¯E⋅B¯+Re⁡∇⋅F=0

In addition, the relationship between the energy density and flux is
(18)ω2Im⁡D⋅E¯+B⋅H¯+∇⋅Savg=0

On the other hand, the local rate of hot-electron generation is defined as [10,11,12]
(19)$${\mathrm{Rate}}_{\mathrm{HE}}\left(r\right)=\frac{2}{\pi^{2}}\frac{e^{2}E_{F}^{2}}{\hbar }\frac{\hbar \omega -\Delta E_{b}}{{\left(\hbar \omega \right)}^{4}}{\left|E_{\mathrm{normal}}\left(r\right)\right|}^{2}$$
where ℏ*ω*, *E_F_* and Δ*E_b_* are the energy of the photon, the Fermi energy of the metal, and the energy barrier between the acceptor state and the Fermi energy of the metal, respectively. Obviously, the local rate of hot-electron generation is related to the intensity of the electric field that is normal for the scatterer’s surface. Using the streamlines of these fluxes and the local intensity of the electric field, we can further investigate the plasmon-enhanced photochemical reaction in the proximity of various nanostructures.

Our method can simulate the EM field of light interacting not only with multiple scatterers but also with a multi-layered scatterer, such as a coreshelled nanoparticle (NP). For example, the efficiencies of the absorption cross-section (ACS) and scattering cross-section (SCS) and the streamline of the energy flux for a spherical coreshell, Au NP-coated with a TiO_2_ layer or Ag NP-coated with an Si layer, were calculated using Mie theory, the MMP method and BEM/MoM, as shown in Appendix A (Appendix A) [42,43]. Additionally, the light-scattering and absorption of a homodimer or heterodimer are studied using the MMP method and BEM/MoM, as shown in Appendix A (Appendix A). Both results are in agreement. Our simulation method is also applicable for investigating the enhancement factor of plasmonic nanostructures in surface-enhanced Raman scattering (SERS), a quantity proportional to the fourth power of an electric field [9].

## 3. Results and Discussion

In the following, we present the numerical results of various plasmonic nanostructures, as irradiated by an upward propagating right-handed (RH) CP light or LP plane wave. The surrounding medium is water with a refractive index of 1.33. The permittivity values for Au and Ag are referenced in Ref. [44]. In the following, the far-field and near-field responses of a nanostructure irradiated by a plane wave are investigated.

### 3.1. Au Nanocube and Nanocuboid

A free-standing Au nanocube (length: 50 nm) in an aqueous solution of Pb(NO_3_)_2_ is irradiated by an upward propagating RH CP light. The efficiencies of ACS and SCS are shown in Figure 2a. Figure 2b shows the streamlines of the energy flux (black) and chirality flux (red) as well as the intensity of the electric field (color) around an Au nanocube, where *λ* = 600 nm. The profiles of the streamlines of the energy flux display helical shapes around the nanocube, as shown in Figure 2b. According to the result of Ref. [1], a helical dielectric cap of PbO_2_ grows initially from a corner at the base and extends upward along the surface of an Au nanocube, as the nanocube is irradiated by a broad-band light with wavelengths longer than 520 nm in an aqueous solution of Pb(NO_3_)_2_ [1]. The helical deposition of PbO_2_ is a result of plasmon-enhanced photo-oxidation via hot carriers (electrons) in the vicinity of the Au nanocube. As a consequence, a composite nanospiroid is formed, which exhibits a chirality of the same handedness with the irradiating CP light. In contrast, the nanocube is an achiral nanostructure. Our result shows that the helical streamline of the energy flux seems like the helical profile of the PbO_2_ cap [1]. Furthermore, the streamlines of the chirality flux create vortex-like patterns winding around these corners, particularly the corners at the bottom. Additionally, the hotspots, the maximum intensity of the electric field, also take place at these corners. Based on these findings, it seems that the initiation of PbO_2_ deposition occurs at any one of the four corners at the bottom randomly, and then the subsequent growth is guided by the energy flux. In fact, the nanocube is placed on the TiO_2_ substrate in the experiment of Ref. [1]. For this photochemical reaction, we propose a plausible hypothesis that the initiation of deposition is determined by the hotspots of the intensity of the electric field, while the growth profile is guided by the streamline of the energy flux. Moreover, the intricate winding behavior of the streamlines of the Poynting vector (energy flux) or chirality flux near the corners may prolong the residence time of hot carriers (electrons) and radicals in the solution, which is a crucial factor for a chemical reaction.

For an Au nanocuboid placed on a TiO_2_ substrate irradiated by an upward propagating RH CP plane wave, the numerical results are shown in Figure 3 [2,13]. The size of the nanocuboid is 110 nm × 40 nm. The corners in this model were rounded with a radius of 10 nm. The refractive index of TiO_2_ is 2.1. The spectra of ACS and SCS are shown in Figure 3a. Figure 3b,c indicate that the profile of the energy flux streamlines exhibits a transversely twisted rolling along the long axis of the nanocuboid at *λ* = 633 nm and 768 nm, respectively. This profile, similar to a twisted donut, is consistent with the experimental results of PbO_2_ deposition on the surface of an Au nanocuboid reported by Ref. [2]. Similar to the nanocube (Figure 2b), the highest intensities of electric field intensity occur at the corners of the bottom, where the streamline of the chirality flux performs like a vortex. However, for the nanocuboid, the helical growth of PbO_2_ is along the transverse axis (long axis), rather than the short axis (optical axis). In contrast, the helical growth of PbO_2_ on the surface of a nanocube is along the optical axis.

### 3.2. Au Nanotriangle and Hexagonal Nanoplate

We analyze an Au nanotriangle or hexagonal nanoplate irradiated by a normal upward incident CP light. The shape of the former is an equilateral triangle with a side length of 100 nm, a thickness of 10 nm and round corners of 3 nm, and that of the latter has a side length of 66.7 nm, a thickness of 10 nm and round corners of 3 nm. The SCS and ACS spectra are shown in Figure 4a and Figure 5a, respectively. Figure 3b and Figure 4b show their streamlines of the energy flux and chirality flux as well as the intensity of the electric field, where *λ* = 875 nm. Of interest, the distribution of the streamlines of the chirality flux looks like a toroidal coil distributed along the periphery of an Au nanotriangle or hexagonal nanoplate, as shown in Figure 4b and Figure 5b. In addition, the distribution of the intensity of the electric field is almost uniform along the edge of the periphery. The phenomenon seems to explain the uniform growth of the Au nanotriangle or hexagonal nanoplate immersed in an aqueous solution of AuHCl_4_ irradiated by a light, which is a photoreduction of Au ions to form Au clusters attached to the periphery of the nanotriangle or hexagonal nanoplate [5].

### 3.3. Au and Ag Bowtie Nanoantennas

We consider an Au bowtie nanoantenna irradiated by CP light (side length: 100 nm, thickness: 10 nm, round corner: 3 nm, gap: 16 nm). In Figure 6a, the ACS and SCS spectra of the Au nanoantenna irradiated by CP light show two peaks at 870 nm and 920 nm corresponding to the SPR induced by LP light with perpendicular or parallel polarization. The two-peak phenomenon in the SCS spectra indicates Rabi splitting energy corresponding to the result of parallel (longitudinal) polarization, separated from the SPR of a single Au nanotriangle (875 nm) [27]. Figure 6b,c show the streamlines of the energy flux (black) and chirality flux (red) as well as the intensity of the electric field (color) induced by CP light of *λ*= 870 nm and 920 nm, respectively. Comparing the top views of the Au bowtie nanoantenna (Appendix A in Appendix A for 870 nm and Figure 6d for 920 nm) with that of the Au nanotriangle (Figure 4c for 875 nm), we found that the electric field and energy flux are confined at the gap zone due to the longitudinal coupling of the SPR of two adjacent and opposing nanotriangles. The maximum of electric field occurs at the tips of bowtie nanoantenna in the gap zone (Figure 6d).

In Figure 7a, the ACS and SCS spectra of the Ag bowtie nanoantenna (side length: 100 nm, thickness: 10 nm, round corner: 3 nm, gap: 16 nm) irradiated by CP light show two peaks of 780 nm and 840 nm corresponding to the SPR induced by LP light with perpendicular or parallel polarization; the Rabi splitting energies of the Ag bowtie nanoantenna are at 1.48 eV (*λ* = 780 nm) and 1.59 eV (*λ* = 840 nm). The spectra versus eV are shown in Appendix A (Appendix A). In contrast, the peak of the ACS spectrum (black) of a single Ag nanotriangle is at 1.55 eV (*λ* = 800 nm), as shown in Figure 7a. Figure 7b,c show the streamlines of the energy flux generated by a 633 nm LP light with the electric field oriented perpendicularly and parallelly to the central line of the nanoantenna, respectively [6]. The winding streamlines of the energy flux appear at the corners of the bowtie (Figure 7b). In contrast, the winding streamlines of the energy flux appear at the tips of the gap instead (Figure 7c). Our results manifest the local confinement of the streamline of the energy flux at the gap or corners, which may illustrate the experimental finding [6]. Note that there is no peak in the neighborhood of *λ* = 633 nm in the ACS and SCS spectra of the bowtie nanoantenna. Hence, the photothermal effect at *λ* = 633 nm is relatively weak. The results induced by a 780 nm LP laser (perpendicular polarization) or 840 nm LP laser (parallel polarization) are also analyzed, as shown in Figure 7d,e, respectively. Both cases correspond to the SPR peaks. The winding streamlines of the energy flux appear at the corners of the bowtie for 780 nm (Figure 7d). In contrast, the winding streamlines of the energy flux appear at the tips of the gap for 840 nm (Figure 7e). The top view is shown in Appendix A (Appendix A). Summarily, only the parallel-polarized light can induce the gap mode of a bowtie nanoantenna. Our finding may provide a new insight into the plasmon-enhanced photochemical reaction, except the local hotspots [6,45].

In summary, our analysis establishes a correlation between the energy flux streamline profile and the asymmetrical, inhomogeneous photogrowth deposition profile on a plasmonic achiral nanocrystal, elucidating the incremental progress of photochemical reactions. Furthermore, the electric field hotspot amplifies local site-selective deposition on plasmonic nanostructures.

## 4. Conclusions

The plasmon-enhanced photo-redox around various plasmonic nanostructures (nanocube/nanocuboid, nanotriangle/hexagonal nanoplate, Au/Ag bowtie nanoantenna) was studied. The streamline distributions of the Poynting vector (energy flux) and the chirality flux as well as the intensity of the electric field surrounding these nanostructures were theoretically analyzed by using a combination of BEM and MoM. It is noteworthy that these streamline patterns exhibit wavelength-dependent and polarization-dependent characteristics. For example, helical streamlines of the energy flux encircle an Au nanocube along the optical axis of a CP light, whereas the streamlines exhibit a transversely twisted rolling along the long axis of the Au nanocuboid perpendicular to the optical axis. In particular, the intricately winding behaviors of these fluxes imply a highly twisted EM field at the gap and corners of the plasmonic bowtie nanoantenna. The distribution of the streamline of the Poynting vector around a bowtie nanoantenna is different from that around a single nanotriangle. For the plasmon-enhanced photochemical redox, we propose a hypothesis that the photo-redox deposition on a plasmonic nanostructure is initiated at the hotspots of the electric field, and the growth profile may be guided by the streamline of the energy flux in the near field. Our findings may provide insight into the mechanism of several previous experiments of plasmon-enhanced photoredox [1,2,3,4,5,6,7,8]. These intriguing phenomena hold the potential to enhance the generation of hot electrons in the proximity of plasmonic nanostructures, thereby paving the way for applications in chirality-dependent photocatalysis [46]. How to apply the PICS and the local rate of hot-electron generation to simulate the incremental growth of a deposition on a plasmonic nanostructure induced by a CP light through the photochemical reaction is an intriguing challenge, which will be useful in manipulating the photochemical process for designing a chiral plasmonic nanocomposite [1,2,12,46].

## Figures and Tables

**Figure 1 nanomaterials-14-00061-f001:**
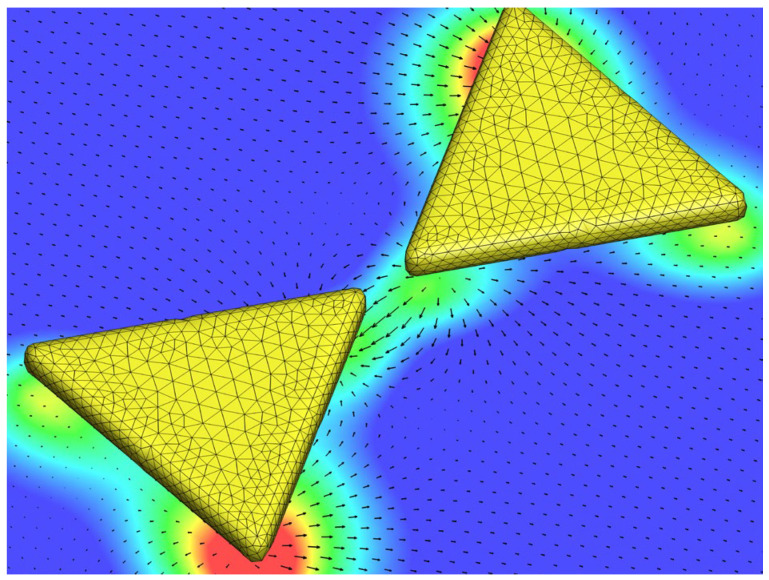
Configuration of implementing BEM/MoM by discretizing the surface of a scatterer (e.g., bowtie nanoantenna) with triangular meshes for dealing with SIEs. Once the surface electric/magnetic charges and currents are obtained, the EM field in the surrounding medium can be calculated from SIRs. For example, the real part and absolute value of the electric field in the surrounding medium are depicted by the vector arrow and color map.

**Figure 2 nanomaterials-14-00061-f002:**
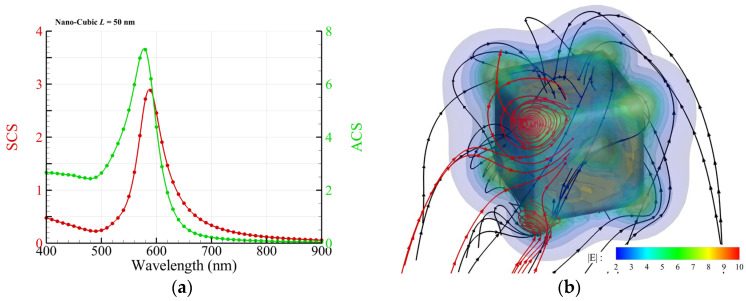
Free-standing Au nanocube with a side length of 50 nm and sharp corners irradiated by an upward propagating RH CP plane wave. (**a**) Spectra of ACS and SCS. (**b**) Streamlines of energy flux (black) and chirality flux (red) as well as the intensity of the electric field $\left|E\right|$ (color) around the Au nanocube, where *λ* = 600 nm. Herein, only the chirality fluxes around two corners are plotted.

**Figure 3 nanomaterials-14-00061-f003:**
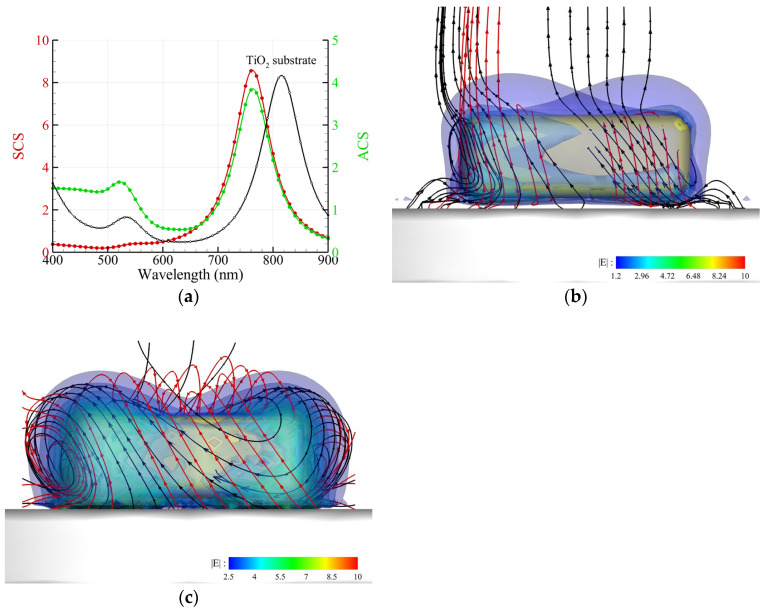
Au nanocuboid with a size of 110 nm × 40 nm and round corners placed on a TiO_2_ substrate irradiated by an upward propagating RH CP plane wave. (**a**) Spectra of the ACS and SCS of a free-standing nanocuboid and the ACS of a nanocuboid on a TiO_2_ substrate (black). (**b**,**c**) Streamlines of the energy flux (black) and chirality flux (red) as well as the intensity of the electric field $\left|E\right|$ (color) around an Au nanocuboid for *λ* = 633 nm and 768 nm, respectively.

**Figure 4 nanomaterials-14-00061-f004:**
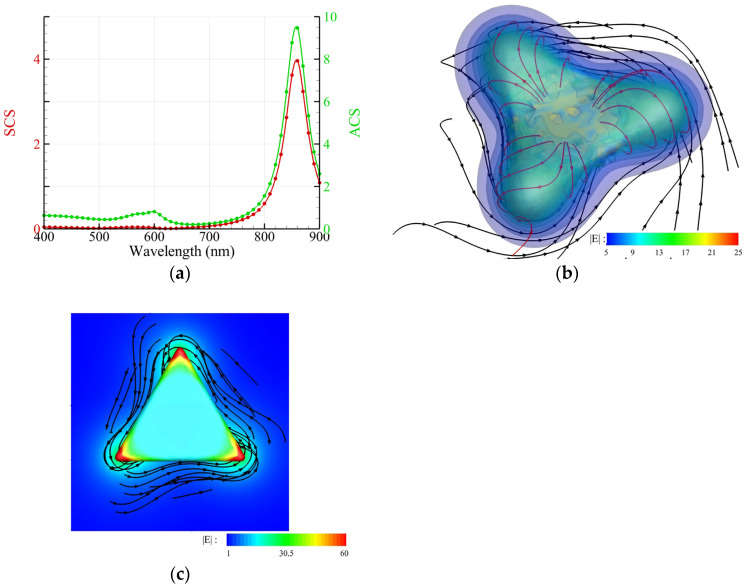
Au nanotriangle (side length: 100 nm, thickness: 10 nm, round corner: 3 nm) irradiated by an upward propagating RH CP plane wave. (**a**) Spectra of SCS and ACS. (**b**) Streamlines of the energy flux (black) and chirality flux (red), as well as the intensity of the electric field $\left|E\right|$ (color) around an Au nanotriangle at *λ* = 875 nm. (**c**) Top view of (**b**).

**Figure 5 nanomaterials-14-00061-f005:**
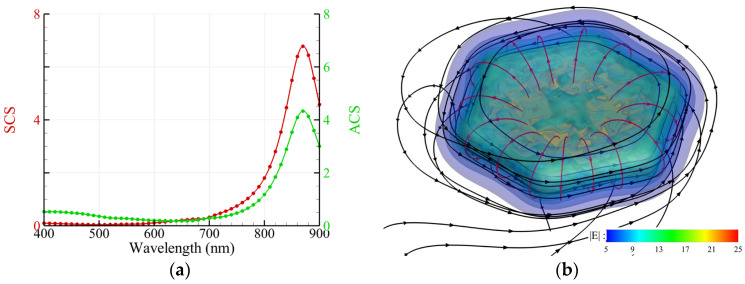
Au hexagonal nanoplate (side length: 66.7 nm, thickness: 10 nm, round corner: 3 nm) irradiated by an upward propagating RH CP plane wave. (**a**) Spectra of SCS and ACS. (**b**) Streamlines of the energy flux (black) and chirality flux (red) as well as the intensity of the electric field $\left|E\right|$ (color) around an Au hexagonal nanoplate at *λ* = 875 nm.

**Figure 6 nanomaterials-14-00061-f006:**
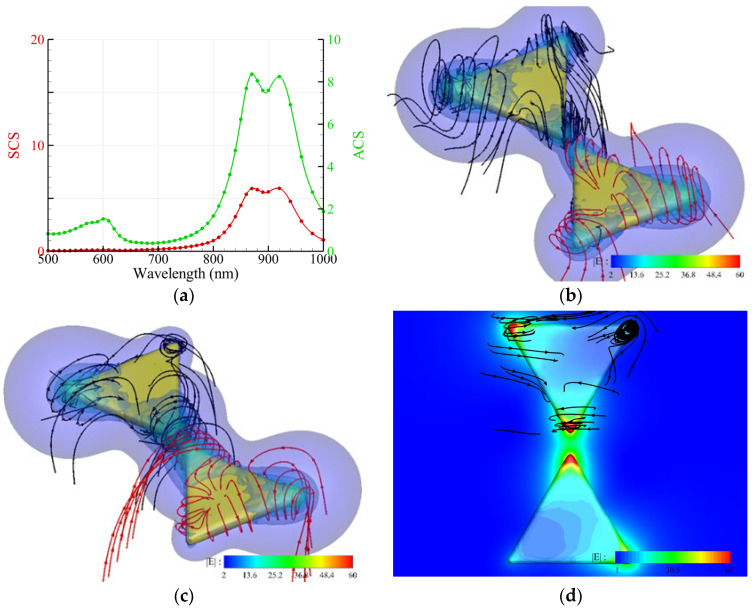
Free-standing Au bowtie nanoantenna in water (length: 100 nm, thickness: 10 nm, rounded corner: 3 nm, gap: 16 nm.) irradiated by an upward propagating RH CP light. (**a**) Spectra of ACS and SCS irradiated by CP light. Streamlines of the energy flux (black) and chirality flux (red) as well as the intensity of the electric field $\left|E\right|$ (color) induced by CP light of (**b**) 870 nm and (**c**) 920 nm, respectively. (**d**) The top view of (**c**).

**Figure 7 nanomaterials-14-00061-f007:**
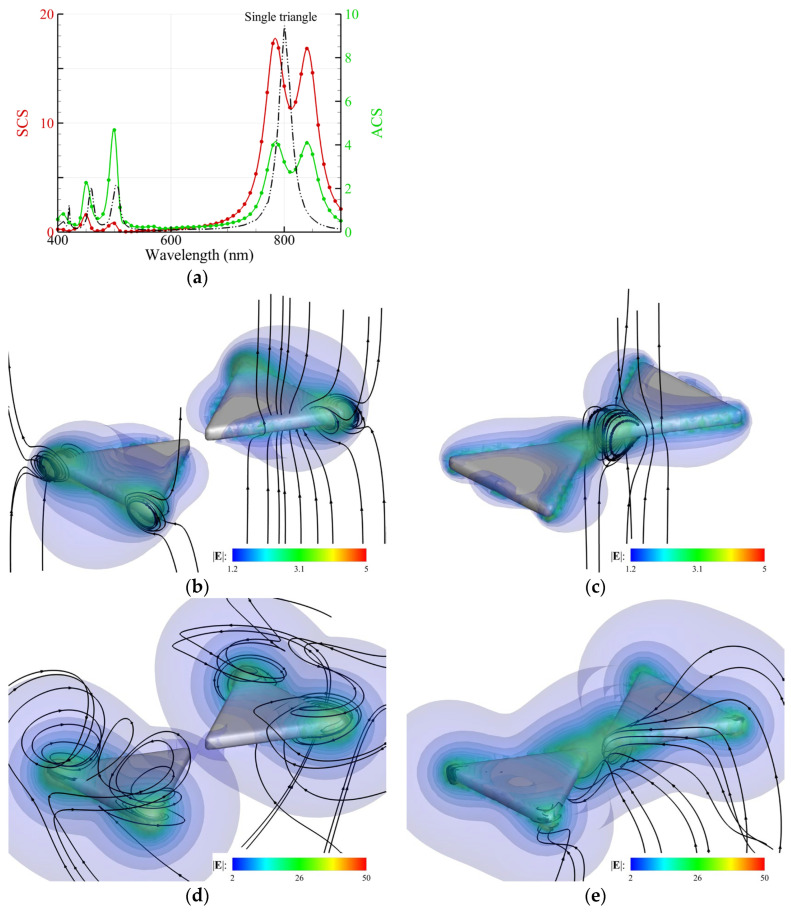
Free-standing Ag bowtie nanoantenna in water (side length: 100 nm, thickness: 10 nm, rounded corner: 3 nm, gap: 16 nm) irradiated by an upward propagating CP or LP light. (**a**) ACS and SCS spectra of the nanoantenna and ACS spectrum of a single triangle (black) induced by CP light. (**b**,**c**) Streamlines of the energy flux (black) and the intensity of the electric field $\left|E\right|$ (color) induced by a 633 nm LP light with perpendicular and parallel polarization with respect to the central line of the nanoantenna, respectively. The results were induced by (**d**) 780 nm LP light (perpendicular) and (**e**) 840 nm LP light (parallel).

## Data Availability

Data are contained within the article and Appendix A.

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
