# Peer review of "Streamlines of the Poynting Vector and Chirality Flux around a Plasmonic Bowtie Nanoantenna"

_nanomaterials, 2023, doi:10.3390/nano14010061_

Round 1

Reviewer 1 Report

Comments and Suggestions for Authors

In the article "Streamlines of Poynting Vector and Chirality Flux around Plasmonic Bowtie Nanoantenna," the authors study the energy and chirality flux densities around different plasmonic nanoantennas with the aim of explaining the morphology of certain photochemical depositions around these structures.

 While the idea is original, from my point of view, there is no evidence in this article of a connection between these two concepts.

There are several reasons for this:

1- The description of flow densities is very succinct in the figures, difficult to understand, and purely descriptive.

2- At no point is the link established with photochemical deposition generation.

I don't think this article is worth publishing as it is.

Comments on the Quality of English Language

English is fine

Author Response

Our responses to these useful comments of reviewer are as follows.

Reviewer I

In the article "Streamlines of Poynting Vector and Chirality Flux around Plasmonic Bowtie Nanoantenna," the authors study the energy and chirality flux densities around different plasmonic nanoantennas with the aim of explaining the morphology of certain photochemical depositions around these structures. While the idea is original, from my point of view, there is no evidence in this article of a connection between these two concepts. There are several reasons for this:

  1. The description of flow densities is very succinct in the figures, difficult to understand, and purely descriptive.

Response: We provide more interpretation with physical meaning for these results.

“Our simulation method is also applicable for investigating the enhancement factor of plasmonic nanostructures in surface-enhanced Raman scattering (SERS), a quantity proportional to the fourth power of electric field.”

“…the Rabi splitting energies of Ag bowtie nanoantenna are at 1.48 eV (l= 780 nm) and 1.59 eV (l= 840 nm). In contrast, the peak of the ACS spectrum (black) of a single Ag nanotriangle is at 1.55 eV (l= 800 nm), as shown in Figure 7a.”

“Comparing the top views of Au bowtie nanoantenna (Figure S5 for 870 nm and Figure 6d for 920 nm) with that of Au nanotriangle (Figure 4c for 875 nm), we found that the electric field and energy flux are confined at the gap zone due to the longitudinal coupling of surface plasmon resonance of two adjacent and opposing nanotriangles.”

  1. At no point is the link established with photochemical deposition generation.

Response: We think that the winding behavior of streamlines of Poynting vector (energy flux) or chirality flux might prolong the residence time of radicals and hot carriers (electrons) for chemical reaction, which is crucial. Our analysis establishes a connection between the energy-flux streamline profile and the asymmetric and inhomogeneous photogrowth deposition profile on a plasmonic achiral nanocrystal and the incremental growth progress of photochemical reactions. In addition, the hotspot of electric field boosts the local site-selective deposition on a plasmonic nanostructure. To illustrate that, several paragraphs are added in the Results and Discussion.

“Moreover, the intricate winding behavior of streamlines of Poynting vector (energy flux) or chirality flux near the corners may prolong the residence time of hot carriers (electrons) and radicals in the solution, which is a crucial factor for chemical reaction.”

“Our results manifest the local confinement of the streamline of energy flux at the gap or corners, which may illustrate the experimental finding [6]. Note that there is no peak in the neighborhood of l= 633 nm in the ACS and SCS spectra of bowtie nanoantenna. Hence, the photothermal effect at l= 633 nm is relatively weak.”

“In summary, our analysis establishes a correlation between the energy-flux streamline profile and the asymmetrical, inhomogeneous photogrowth deposition profile on a plasmonic achiral nanocrystal, elucidating the incremental progress of photochemical reactions. Furthermore, the electric field hotspot amplifies local site-selective deposition on plasmonic nanostructures.”

The attached manuscript has been revised and Supplementary Materials is added according to the useful comments of reviewers. If there is any other question, please do not hesitate to inform us. Thanks for your help.

Sincerely

Jiunn-Woei Liaw

Professor

Department of Mechanical Engineering, Chang Gung University, Taiwan

2023/11/23

Reviewer 2 Report

Comments and Suggestions for Authors

Comments on the Quality of English Language

Author Response

Reviewer II

The results are interesting and the article is well-written. I support the publication of the article after the following questions are addressed:

  1. Can the authors comment about the possibility of using the described method for analyzing dielectric nanoparticles instead of plasmonic ones?

Response: Our BEM+MoM is more suitable for the simulation of dielectric nanostructure than plasmonic nanostructure. For example, the results of coreshell (Au NP coated with TiO2 layer), which belongs to multi-layered structures, calculated by BEM+MoM are provide in Figure S1 (Supplementary Materials), and the corresponding results of Mie theory are also provided for verification. The numbers of the surface triangular mesh for BEM+MoM are 864 for the discretization of the outer interface (water/TiO2) and the inner interface (TiO2/Au), respectively. Both results are consistent. However, the accuracy of BEM+MoM at the short-wavelength region can be improved by raising the number of triangular mesh.

Figure S1. The spectra of efficiencies of SCS and ACS of TiO2@Au NP in water. Solid line: Mie theory. Dash line with circle: BEM+MoM. The radius of Au NP is 50 nm, and the thickness of TiO2 layer is 100 nm.

  1. How would the method be affected by considering hybrid nanostructures, for example a nanoparticle dimer where one particle is made of gold and the other one is made of silicon?

Response: Our BEM+MoM also can simulate the hybrid nanostructures, such as a heterogeneous dimer, Au NP with Si NP. The results are provided in Figure S4 (Supplementary Materials), and the corresponding results of MMP are also provided for verification. Both results are consistent. The following figure demonstrate that. In the revised manuscript, a paragraph is added to illustrate that.

“Our method can simulate the EM field of light interacting not only with multiple scatterers but also with multi-layered scatterer, such as a coreshelled nanoparticle (NP). For example, the efficiencies of absorption cross section (ACS) and scattering cross section (SCS) and streamline of energy flux for a spherical coreshell, Au NP coated with a TiO2 layer or Ag NP coated with a Si layer, were calculated by using Mie theory, MMP method and BEM+MoM, as shown in Figures S1 and S2 (Supplementary Materials). Additionally, the light scattering and absorption of a homodimer or heterodimer are studied by using MMP method and BEM+MoM, as shown in Figures S3 and S4 (Supplementary Materials). Both results are in agreement.”

Figure S4. The spectra of efficiencies of SCS and ACS of a heterogeneous dimer, Au NP and Si NP along x axis, irradiated by circularly polarized plane wave in water. The radii of these NPs is 50 nm and the gap is 30 nm. Solid line: MMP. Circle symbol: BEM+MoM with 484 meshes on each NP.

  1. More novel configuration such as NP on mirror configuration have demonstrated extremely high confinement of electromagnetic radiation in small mode volumes. Can the authors comment on the Poynting flux in those configurations and the possibility of being used for photocatalytic applications?

Response: The local surface plasmon resonance (LSPR) of bowtie nanoantenna is a high confinement behavior of the EM energy at the gap zone. For example, by comparing the distributions of the intensity of electric field and the energy flux of a single Au nanotriangle at the SPR mode (875 nm) with those of the bowtie nanoantenna (a pair of opposing nanotriangles) at SPR modes of Rabi splitting (870 nm and 920 nm), we found that the local confinement of EM energy and energy flux in the gap zone is obvious, as shown in the following figures. In particular, the phenomenon is more severs, as the longitudinal coupling of the two opposing nanotriangles at 920 nm is induced; this is majorly induced by the parallel-polarized component of the incident light, r.w.t. the longitudinal axis of the bowtie nanoantenna. This phenomenon could be utilized for photocatalytic application.  

(a)

(b) (c)

Figure. (a) Electric fields and streamlines of energy flux of a single Au nanotriangle irradiated by CP light of 875 nm. The results of Au bowtie nanoantenna at (b) 870 nm, and (c) 920 nm (Top view).

  1. Core-shell nanoparticles are of great interest due to the coupling of the modes in the core with the modes in the shell, as well as their integration into different environments (as the shell can act as a protective layer). Can the authors include the analysis of this geometry in the manuscript? In this regard, I recommend to include the following reference, where a deep analysis of scattering by means core-shell nanoparticles is performed: [1] Ángela I Barreda et al 2016 Nanotechnology 27, 234002.

Response: This useful reference is cited in the revised manuscript. To study the local energy confinement and the streamline of energy flux (Poynting vector) around a coreshell, we used MMP method. We add a paragraph in Supplementary Materials. For example, Ag NP (r= 70 nm) is the core coated with a Si layer (thickness: 160 nm) irradiated by a x-polarized plane wave upward propagating along z axis [Nanotechnology 2016, 27 234002]. The wavelength-dependent dielectric constant of Si is referred to Ref. [Palik, E. D. Handbook of Optical Constants of Solids]. Figure S2a shows the spectra of efficiencies of SCS and ACS of coreshell in water. The electromagnetic distributions in nearfield at two peaks of ACS, 938 nm and 1187 nm, are analyzed. Figures S2b and S2c show the maps of streamlines of energy flux and  in the x-z plane and y-z plane cross sections, respectively, at l= 938 nm. Figures S2d and S2e show the results in the x-z plane and y-z plane cross sections, respectively, at l= 1187 nm. The local energy confinement and the intricate vortex of energy-flux streamline are observed in these figures.

(a)

(b) (c)

(d) (e)

Figure S2. (a) The spectra of efficiencies of SCS and ACS of a coreshelled NP in water, irradiated by a x-polarized plane wave upward propagating along z axis; Ag NP of r= 70 nm is coated with Si layer of 160-nm thickness. Maps of streamlines of energy flux and  of (b) x-z plane and (c) y-z plane cross sections at l= 938 nm. Maps of streamlines of energy flux and  of (d) x-z plane and (e) y-z plane cross sections at l= 1187 nm. Color bar:

  1. How useful is the described method for SERS enhancement?

Response: The enhancement factor for SERS is proportional to the fourth power of electric field. We add a sentence to illustrate this application on SERS in the Method.

“Our simulation method is also applicable for investigating the enhancement factor of plasmonic nanostructures in surface-enhanced Raman scattering (SERS), a quantity proportional to the fourth power of electric field.”

  1. I recommend to carefully read the manuscript as there are a few typos. For example, scatter in fin line 168.

Response: The typo of “scatter” is replaced by “scatterer”.

  1. Which is the values of the Rabi splitting in energy?

Response: For a single Ag nanotriangle in water, the peak of the ACS spectrum (black) is at 1.55 eV (l= 800 nm), as shown in the following Figure. The Rabi splitting energies of Ag bowtie nanoantenna are at 1.48 eV (l= 780 nm) and 1.59 eV (l= 840 nm). To illustrate that, two sentences are added in the Results and Discussion.

“…the Rabi splitting energies of Ag bowtie nanoantenna are at 1.48 eV (l= 780 nm) and 1.59 eV (l= 840 nm). In contrast, the peak of the ACS spectrum (black) of a single Ag nanotriangle is at 1.55 eV (l= 800 nm), as shown in Figure 7a.”

The attached manuscript has been revised and Supplementary Materials is added according to the useful comments of reviewers. If there is any other question, please do not hesitate to inform us. Thanks for your help.

Sincerely

Jiunn-Woei Liaw

Professor

Department of Mechanical Engineering, Chang Gung University, Taiwan

2023/11/23

Reviewer 3 Report

Comments and Suggestions for Authors

The paper describes the numerical study of Poynting vector and chhirality flux around various plasmonic nanostructures - nanocubes, nanocuboids, nanotriangles, hexagonal nanoplates and bowtie nanoantennas. The results shown in the manuscript were obtained either for circularly polarized or linearly polarized light. The numerical modeling was performed by means of boundary element method combined with the method of moment.
The manuscript is well structured and the quality of figures is satisfactory. The supplementary materials are an important addition to the manuscript, as they provide insight into the boundary element method combined with the method of moment applied to different structures.

Comments on the Quality of English Language

The manuscript needs revision for language and grammar, as there are some typos and incomprehensible sentences - e.g. in lines 218, 239-240, 275.

Author Response

The manuscript needs revision for language and grammar, as there are some typos and incomprehensible sentences - e.g. in lines 218, 239-240, 275.

Response: These typos and sentences were corrected. The English writing was revised.

Reviewer 4 Report

Comments and Suggestions for Authors

The authors theoretically analyzed the intensity of the electric field around the nanostructure, as well as the streamline distribution of the pointing vector (energy flux) and chirality flux, using a combination of BEM and MoM. This manuscript is qualified for publication in this journal, but they need to address and revise followings.

1. The authors conducted their simulations assuming an aqueous solution of Pb(NO3)2. Does the acidity of the solution affect the SCS and ACS or photoredox? Typically, many synthesis processes takes place in somewhat harsh environment. And there are byproduct particles that are created throughout the solution, will these have an impact? I know this is a theoretical work, but I think the authors need to consider real experimental environments and understand the applicability of their simulations.

2. Authors often use "BEM+MoM". Is this a universally used expression? If not, it would be better to avoid using the math operator "+" in a non-strict sense.

Comments on the Quality of English Language

The manuscript is written in a way that makes it easy to read.

Author Response

  1. The authors conducted their simulations assuming an aqueous solution of Pb(NO3)2. Does the acidity of the solution affect the SCS and ACS or photoredox? Typically, many synthesis processes take place in somewhat harsh environment. And there are byproduct particles that are created throughout the solution, will these have an impact? I know this is a theoretical work, but I think the authors need to consider real experimental environments and understand the applicability of their simulations.

Response: Since the concentration of Pb(NO3)2 in the photochemical reaction of Ref. [1,2] was very low, we assume that the refractive index of aqueous solution is the same with pure water for the simulation of light-matter interaction, including SCS and ACS spectra. However, once the PbO2 layer (product), which is a high-k dielectric material, is generated from the photo-oxidation and deposited on the gold nanostructure, the ACS and SCS of the new-growth composite nanostructure will be changed significantly; normally the red-shift will be induced. Therefore, the effective refractive index of a byproduct of the photochemical reaction, which could be deposited on a plasmonic nanostructure, should be considered. In this paper, we did not take this new-growth dielectric layer into account. Indeed, the issue of studying the performance of heterogeneous nanostructures will be important in the future. In Conclusion, we add a sentence to highlight the future work of simulation.   

“How to apply the PICS and the local rate of hot-electron generation to simulate the incremental growth of a deposition on a plasmonic nanostructure induced a CP light through the photochemical reaction is an intriguing challenge, which will be useful in manipulating the photochemical process for designing a chiral plasmonic nanocomposite [1, 2, 12, 46].”

  1. Authors often use "BEM+MoM". Is this a universally used expression? If not, it would be better to avoid using the math operator "+" in a non-strict sense.

Response: We use “BEM/MoM” to replace “BEM+MoM” in the revised manuscript. 

Round 2

Reviewer 2 Report

Comments and Suggestions for Authors

I recommend the article for publication 

Author Response

Thanks.